# Phages ZC01 and ZC03 require type-IV pilus for *Pseudomonas aeruginosa* infection and have a potential for therapeutic applications

Layla Farage Martins,[1,2] Ariosvaldo Pereira dos Santos Junior,[1,2] Gianlucca Gonçalves Nicastro,[1] Gaby Scheunemann,[3] Claudia Blanes Angeli,[4] Fernando Pacheco Nobre Rossi,[1] Ronaldo Bento Quaggio,[1] Giuseppe Palmisano,[4,5] Germán Gustavo Sgro,[2,6] Kelly Ishida,[3] Regina Lúcia Baldini,[1,2] Aline Maria da Silva[1,2]

**ABSTRACT**    There has been a growing interest in bacteriophages as therapeutic agents to treat multidrug-resistant bacterial infections. The present work aimed at expanding the microbiological and molecular characterization of lytic phages ZC01 and ZC03 and investigating their efficacy in the control of *Pseudomonas aeruginosa* infection in an invertebrate animal model. These two phages were previously isolated from composting using *P. aeruginosa* strain PA14 as the enrichment host and had their genomes sequenced. ZC01 and ZC03 present, respectively, siphovirus and podovirus morphotypes. ZC01 was recently classified into the genus *Abidjanvirus*, while ZC03 belongs to *Zicotriavirus* genus of the *Schitoviridae* N4-like viruses. Through proteomics analysis, we identified virion structural proteins of ZC01 and ZC03, including a large virion-associated RNA polymerase that is characteristic of N4-like viruses, some hypothetical proteins whose annotation should be changed to virion structural proteins and a putative peptidoglycan hydrolase. Phages ZC01 and ZC03 exhibit a limited yet distinct host range, with moderate to high efficiency of plating (EOP) values observed for a few *P. aeruginosa* clinical isolates. Phage susceptibility assays in PA14 mutant strains point to the type-IV pilus (T4P) as the primary receptor for phages ZC01 and ZC03, and the major pilin (PilA$_{PA14}$) is the T4P component recognized by these phages. Moreover, both phages significantly increase survival of *Galleria mellonella* larvae infected with PA14 strain. Taken together, these results underpin the therapeutic potential of these phages to treat infections by *P. aeruginosa* and lay the groundwork for a more detailed investigation of phage-bacteria-specific recognition mechanisms.

**IMPORTANCE**    Phage therapy is gaining increasing interest in cases of difficult-to-treat bacterial human infections, such as carbapenem-resistant *Pseudomonas aeruginosa*. In this work, we investigated the molecular mechanism underlying the interaction of the lytic phages ZC01 and ZC03 with the highly virulent *P. aeruginosa* PA14 strain and their efficacy to treat PA14 infection in *Galleria mellonella* larvae, a commonly used invertebrate model for phage therapy. We depicted the protein composition of ZC01 and ZC03 viral particles and identified pilin A, the major component of type-4 pilus, as the receptor recognized by these phages. Our findings indicate that phages ZC01 and ZC03 may be further used for developing therapies to treat multidrug-resistant *P. aeruginosa* infections.

**KEYWORDS**    *Pseudomonas aeruginosa*, PA14, phage therapy, phage receptor, type-4 pilus

Address correspondence to Layla Farage Martins, layla@iq.usp.br.

The authors declare no conflict of interest.

See the funding table on p. 14.

Bacteriophages (phages) are viruses that rely on a bacterial host for propagation. Phages are mainly classified according to their life cycle into lysogenic and lytic, in the latter of which the host cells are lysed and mature phage particles (virions) are released (1). Phages generally display high specificity toward bacterial species or strains, and this is determined by the mechanisms of phage adsorption to host cells (2) and by bacterial antiphage defense systems (3).

The receptors known to be involved in phage adsorption to Gram-negative bacteria are lipopolysaccharides (LPS), capsular polysaccharides, pili, flagella, and outer membrane proteins (2, 4, 5). Phages, on the other hand, have receptor-binding proteins (RBPs) responsible for the specific recognition and interaction with the receptor displayed on the surface of bacterial cells, thus initiating the infection process (2). As the first point of contact with bacterial cells, RBPs are the primary determinants of the phage ability to infect one or more bacterial strains, an attribute referred to as phage host range. RBPs are typically located on tail fibers or tailspikes which may have enzymatic activity that binds and degrades carbohydrate moieties on the bacterial surface (6–8).

Similarly to other bacterial species, the major receptors that have been implicated in phage adsorption to *Pseudomonas aeruginosa* are type IV pili (T4P) (9–12) and LPS (13–17). Spontaneous mutations in genes responsible for T4P or LPS synthesis can result in phage-resistant mutants, particularly under laboratory conditions (17–20). On the other hand, virulence reduction of phage-resistant bacteria may occur if the phage targets a virulence factor (5, 15, 21, 22).

*P. aeruginosa* belongs to the ESKAPEE group of pathogens (*Enterococcus faecium*, *Staphylococcus aureus*, *Klebsiella pneumoniae*, *Acinetobacter baumannii*, *P. aeruginosa*, *Enterobacter* spp., *and Escherichia coli*) which became a serious concern due to the worldwide antimicrobial resistance (AMR) increase in nosocomial and community-acquired infections (23–27). Multidrug-resistant *P. aeruginosa* causes acute or chronic infection in immunocompromised individuals with chronic obstructive pulmonary disease, cystic fibrosis, cancer, traumas, burns, sepsis, and ventilator-associated pneumonia (28). In addition to the development of new antimicrobials to combat AMR (29), phage therapy is gaining increasing interest in cases of difficult-to-treat ESKAPEE human infections (30–33), such as carbapenem-resistant *P. aeruginosa* (34–36).

PAO1 and PA14 (or UCBPP-PA14) are commonly used as laboratory reference *P. aeruginosa* strains to study this bacterial species (37–40). While our PAO1 is a derivative of the original PAO1 isolate which was obtained from a wound at Melbourne, PA14 was one of the strains isolated from burn wound patients at a hospital in Pennsylvania (37, 38, 41). PA14 has been shown to be highly virulent in both animals and plants (42, 43), and its use in research is gradually matching that of *P. aeruginosa* PAO1. Until now, thousands of phages that infect *P. aeruginosa* PAO1, PA14, and/or clinical strains have been characterized, and some of them exhibit a broad spectrum of activity against *P. aeruginosa* clinical isolates (34, 44–49). Nevertheless, new *P. aeruginosa* phages are continuously being discovered and characterized through phage isolation studies and predictions retrieved from metagenomics data set (50, 51).

The *Pseudomonas* phages ZC01 and ZC03 were isolated from composting samples using PA14 strain as the enrichment host and previously classified as *Siphoviridae* and *Podoviridae*, respectively, based on their double-strand DNA (dsDNA) genomes and morphotypes of tailed phages (46). Phage ZC01 was recently classified into the genus *Abidjanvirus*, while ZC03 is a single species of the new *Zicotriavirus* genus (52, 53). These two phages had their genomes analyzed, but their biological properties were not fully explored yet. In this work, we extended the molecular characterization of phages ZC01 and ZC03 and investigated the mechanism underlying their interaction with *P. aeruginosa*. Both phages were evaluated regarding their phage therapy potential against PA14 infection in *Galleria mellonella* larvae. Our findings indicate that these phages may be further used for developing therapies to treat multidrug-resistant *P. aeruginosa* infections.

## RESULTS

### Update of ZC01 and ZC03 phage characteristics

The phages ZC01 and ZC03 were isolated from composting samples using *P. aeruginosa* strain PA14 as enrichment host and exhibit, respectively, siphovirus and podovirus morphotypes (46) (Fig. 1). Based on their complete genome sequences (sequence accessions NC_052965 and NC_048638, these phages are currently classified as *Caudoviricetes; Mesyanzhinovviridae; Bradleyvirinae; Abidjanvirus; Abidjanvirus ZC01* and *Caudoviricetes; Schitoviridae; Zicotriavirus; Zicotriavirus ZC03* according to the new taxonomy of bacterial viruses that abolished the morphology-based families *Myoviridae*, *Siphoviridae*, and *Podoviridae* (52). ZC01 genome is highly similar (93% coverage and 96% identity) to the genome of phage Ab18 (54). While Ab18 infects *P. aeruginosa* strain PAO1 but not PA14 (54), ZC01 infects PA14 but not PAO1 (46). ZC03 and phage ZC08 (46) are the only members of the genus *Zicotriavirus* of the *Schitoviridae* N4-like viruses that encode three RNA polymerase genes, including a large (~3,500 aa) virion-associated RNA polymerase that is characteristic of this family (55).

Both phages form clear lysis plaques, which is typical for lytic phages. ZC03 presents a latent period of ~50 min and a calculated burst size of 10 phage particles per infected cell (46). On the of other hand, ZC01 presents a latent period of ~100 min and a calculated burst size of 87 phage particles per infected cell (Fig. S1). The bacteriolytic effect of phages ZC01 and ZC03 on PA14 strain was investigated through time-killing curves at different multiplicities of infection (MOIs). A better lytic effect was observed

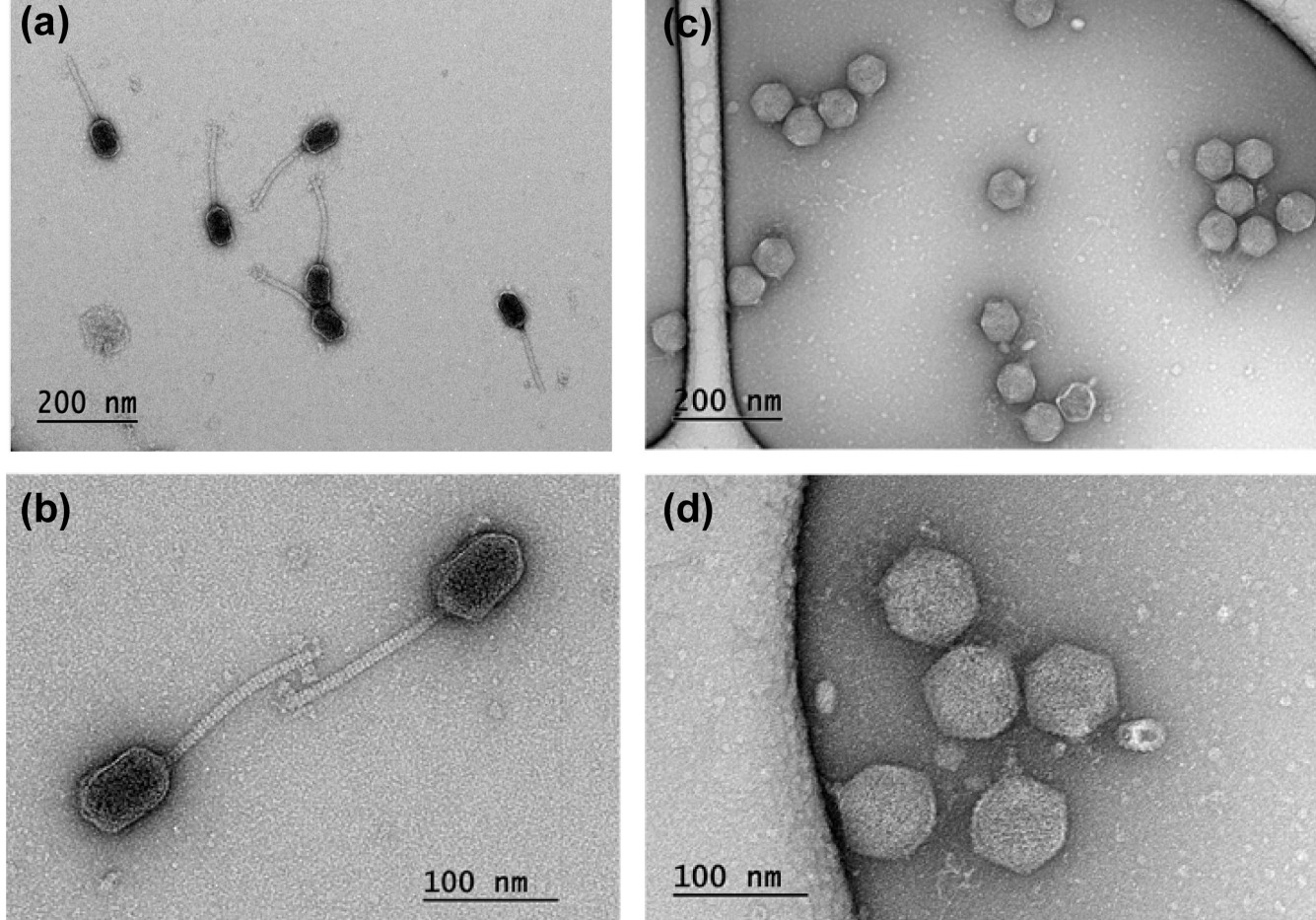

**FIG 1** Transmission electron micrographs of ZC01 and ZC03 purified phage particles, negatively stained with uranyl acetate. Different areas on a grid show intact ZC01 (a and b) and ZC03 (c and d) phage particles with siphovirus (ZC01) and podovirus (ZC03) morphotypes as previously described (46).

at MOIs higher than 1 for both phages (Fig. S2). It is worth noting that until 12 h, no secondary bacterial growth was detected upon ZC01 and ZC03 infection, indicating a prolonged control of bacterial growth under these conditions, with no apparent phage resistance emergence.

Phage ZC01 was stable at 25°C and 37°C, but its viability decreased upon incubation at 16°C, 42°C, and 60°C. In contrast, phage ZC03 was moderately stable at 16°C and 42°C, but at 60°C, its titer was drastically reduced. Both phages were fully inactivated by incubation at 80°C or when exposed to UV light for 20 min. ZC01 and ZC03 were reasonably stable in chloroform 10% and in pH values ranging from 4 to 12. Both phages became unviable at pH 2. These results are summarized in Table 1.

## Proteomics of ZC01 and ZC03 phage particles

Mass spectrometry-based proteomics of highly pure ZC01 and ZC03 phage particles (Fig. 1) identified 51 proteins out of 78 predicted open reading frames (ORFs) in the ZC01 genome (Table S1) and 65 proteins out of 85 predicted ORFs in the ZC03 genome (Table S2). For both phage particles, the in-gel trypsin digestion outperformed the in-solution trypsin digestion (shotgun proteomics) in terms of the identification of higher number of polypeptides. Table 2 lists 18 proteins of ZC01 virions which had a coverage higher than 20% and a minimum of three peptides (Table S1). The major head protein (ZC01_055) and the tail length tape measure protein (ZC01_065) had the highest coverage and number of peptides. Other typical structural proteins, such as portal protein (ZC01_047), head morphogenesis protein (ZC01_048), tail fiber assembly protein (ZC01_068 and ZC01_069), and tail protein (ZC01_072), are among the proteins that have been reliably identified in ZC01 virions.

For ZC03, 24 proteins (Table 2) had a coverage higher than 20% and a minimum of three peptides (Table S2). Among the predominant proteins identified, in terms of coverage and number of peptides, are the large virion-associated RNA polymerase (ZC03_015), typical of the N4-Like viruses (family *Schitoviridae*), which carry this 399.4 kDa enzyme inside their capsids. Other dominant proteins identified are a putative peptidoglycan hydrolase (ZC03_016), the major head protein (ZC03_021), and the portal protein (ZC03_024). Proteomics of ZC03 virion also identified two ORFs predicted as tail fiber proteins (ZC03_005 and ZC03_006).

## ZC01 and ZC03 host-range evaluation

Previously reported drop test assays revealed that phages ZC01 and ZC03 present a narrow host range producing clear lysis plaques in just 3 out of 18 *P. aeruginosa* isolates. Moreover, both phages infect the reference strain PA14 but not strain PAO1 (46). Additional drop test assays were performed using 66 *P. aeruginosa* clinical and environmental isolates (Table S3) some of them with multidrug-resistance phenotypes. While seven isolates were susceptible to both phages, two isolates were susceptible to ZC03 only, and one isolate was susceptible just to ZC01. Altogether, and including the three susceptible isolates previously identified, 11.5% (8/69) of the isolates were susceptible to ZC01 and ZC03, respectively, whereas 16% (11/69) and 14% (10/69) of the *P. aeruginosa*

**TABLE 1** Viability of phages ZC01 and ZC03 under different conditions

| Phage | Temperature | | | | | |
|---|---|---|---|---|---|---|
| | 16°C | 25°C | 37°C | 42°C | 60°C | 80°C |
| ZC01 | 24.4 ± 15 | 103.3 ± 1 | 116.0 ± 22 | 33.9 ± 3 | 30.4 ± 13 | 0.2 ± 1 |
| ZC03 | 89.8 ± 20 | 96.0 ± 2.5 | 89.9 ± 19 | 76.5 ± 26 | 3.9 ± 3.7 | 0.0 |
| phage | pH | | | | | CHCl$_3$ | UV |
| | 2 | 4 | 7.5 | 9 | 12 | 10% | 20 min |
| ZC01 | 0 | 59.6 ± 31 | 99.8 ± 78 | 52.8 ± 44 | 53.2 ± 31 | 73.4 ± 7 | 0 |
| ZC03 | 0 | 66.4 ± 5 | 117.1 ± 24 | 73.5 ± 12 | 51.8 ± 20 | 72.0 ± 14 | 0 |

[a]Incubations of phage suspension at different temperatures, pHs, and CHCl3 were performed for 60 min. Phage % viability (mean ± SD from three independent assays) was relative to the initial phage titer (109 PFU/mL) in control condition (25°C in Saline-Magnesium buffer pH 7.5) before the test incubations.

**TABLE 2** Proteins of ZC01 and ZC03 virions identified by proteomic analysis which had a coverage higher than 20% and presented a minimum of three peptides[a]

| ZC01 ORF ID | Predicted function | ZC03 ORF ID | Predicted function |
|---|---|---|---|
| ZC01_047 | Portal protein | ZC03_001 | Hypothetical protein |
| ZC01_048 | Head morphogenesis protein | ZC03_002 | Tail assembly protein |
| ZC01_054 | Head scaffolding protein | ZC03_003 | Hypothetical protein |
| ZC01_055 | Major head protein | ZC03_004 | Hypothetical protein |
| ZC01_056 | Hypothetical protein | ZC03_005 | Tail fiber protein |
| ZC01_057 | Virion structural protein | ZC03_006 | Tail fiber protein |
| ZC01_058 | Head-tail adaptor Ad1 | ZC03_008 | Hypothetical protein |
| ZC01_060 | Tail completion or Neck1 protein | ZC03_012 | Single-stranded DNA-binding protein |
| ZC01_061 | Tail terminator protein | ZC03_014 | Hypothetical protein |
| ZC01_062 | Minor tail protein | ZC03_015 | Virion RNA polymerase |
| ZC01_065 | Tail length tape measure protein | ZC03_016 | Peptidoglycan hydrolase |
| ZC01_066 | Structural protein | ZC03_017 | Hypothetical protein |
| ZC01_067 | Structural protein | ZC03_018 | Hypothetical protein |
| ZC01_068 | Tail assembly protein | ZC03_019 | Virion structural protein |
| ZC01_069 | Tail assembly protein | ZC03_021 | Major head protein |
| ZC01_072 | Tail protein | ZC03_022 | Tail length tape measure protein |
| ZC01_073 | Hypothetical protein | ZC03_024 | Portal protein |
| ZC01_075 | Baseplate hub subunit and tail lysozyme | ZC03_028 | Hypothetical protein |
|  |  | ZC03_029 | Hypothetical protein |
|  |  | ZC03_030 | Virion structural protein |
|  |  | ZC03_044 | Hypothetical protein |
|  |  | ZC03_087 | Thymidylate synthase |
|  |  | ZC03_089 | RIIB lysis inhibitor |
|  |  | ZC03_094 | Hypothetical protein |

[a]The complete lists of proteins identified with the respective number of peptides and coverage from in-gel and in-solution proteomics are presented on Tables S1 and S2. ORF amino acid sequences and functional prediction can be found along with ZC01 (accession NC_052965) and ZC03 (accession NC_048638) NCBI Reference Sequence genome annotation.

isolates were susceptible to ZC01 or ZC03. However, the efficiency of plating (EOP) of none of these 13 susceptible clinical isolates has surpassed that of the strain PA14 (Table 3). While for most of these 13 isolates EOP values of ZC01 were moderate (>0.1%), lower EOP values (<0.1%) were calculated for ZC03 (Table 3). In contrast, higher EOP values (~50%) were calculated for the lung isolates Fc79a M and Fc79a PAB NM infected by ZC03.

## Identification of type 4 pilus as the receptor for ZC01 and ZC03

Commonly identified receptors for *P. aeruginosa* phages are LPS and T4P (4). To determine whether T4P or LPS would function as receptors for ZC01 and ZC03, mutants PA14Δ*pilA*, lacking pilin A, and PA14_OAg⁻, lacking the O antigen, were tested on phage infection assays. As shown in Fig. 2a, both phages form clear lysis plaques in strain PA14 and in PA14_OAg⁻, but not in PA14Δ*pilA* strain. As previously observed (46), PAO1 strain is resistant to phages ZC01 or ZC03 (Table S3). Twitching motility, which is dependent on functional T4P, was fully abolished in the PA14Δ*pilA* but was not altered in the other strains, including the PA14_OAg⁻ (Fig. 2b). These findings strongly suggest that T4P is the primary receptor for ZC01 and ZC03 adsorption to PA14 cells.

To verify that pilin A, a major component of T4P in *P. aeruginosa*, is indeed the primary determinant for ZC01 and ZC03 adsorption to PA14 cells, *pilA* of PA14 (*pilA*$_{PA14}$) or *pilA* of PAO1 (*pilA*$_{PAO1}$) was used to complement the PA14Δ*pilA* mutant strain. As shown in Fig. 3a, PA14Δ*pilA* complemented with *pilA*$_{PA14}$, but not *pilA*$_{PAO1}$, can be infected by the

**TABLE 3** EOP of phages ZC01 and ZC03 for selected clinical isolates[a]

| P. aeruginosa strain/isolate | Drop test assay | | EOP | | Source |
|---|---|---|---|---|---|
| | ZC01 | ZC03 | ZC01 | ZC03 | |
| PA14[b] | + | + | 100% | 100% | Reference strain |
| 3845 GSP-3 producer | + | + | 3.4% | 0.02% | HIV patient feces |
| ALERTA 226 (GES-5 carbapenemase-producer) | − | + | nd | 0.06% | Hospital |
| ALERTA 275 (VIM-7 carbapenemase-producer) | − | + | nd | 0.05% | Hospital |
| ALERTA 395 (IMP-18 metallo-beta-lactamase producer) | + | + | 5% | 9% | Hospital |
| Fc79a M | + | + | 2% | 49% | Lung secretion |
| Fc79a PAB NM | + | + | 8% | 51% | Lung secretion |
| Fc7f NM | + | − | 2% | nd | Lung secretion |
| MT222 | + | + | 14% | 0.01% | Tracheal aspirate |
| P13.612 | + | + | 20% | 0.03% | Hospital |
| SC-61 | + | + | 8% | 2% | Nasal secretion |
| H6044[c] | + | − | 0.6% | nd | Blood |
| H6086[c] | + | − | 14% | nd | Blood |
| 5757[c] | + | + | 9.8% | 0.045% | Blood and urine |

[a]Clear lysis plaque (+) and no lysis/turbid plaque (−). The EOP value of 100% was considered for the host strain. nd: not determined.
[b]Host strain.
[c]Isolates analyzed in previous work (46). The full list of the 69 strains evaluated is shown on Table S3.

two phages. This result suggests that pilin A sequence variation may be the primary host barrier for ZC01 and ZC03 susceptibility. PA14Δ*pilA*, when complemented with either *pilA*$_{PA14}$ or *pilA*$_{PAO1}$, exhibits its twitching motility restored, suggesting that T4P is functional upon complementation (Fig. 3b).

To further evaluate the dependence of PA14 pilin A as the receptor for ZC01 and ZC03, *pilA*$_{PA14}$ was introduced in PAO1 (co-expression) and in a PAO1Δ*pilA* mutant (complementation). PilA$_{PA14}$ turns these PAO1 strains susceptible to infection by phage ZC01 but not to ZC03 (Fig. 4a). Twitching motility was restored in the PAO1Δ*pilA* upon complementation, and it was not altered by co-expression of PilA$_{PA14}$ in wild-type PAO1 (Fig. 4b).

## Evaluation of therapeutic potential of phages ZC01 and ZC03 in *G. mellonella*

The therapeutic potential of phages ZC01 and ZC03 against *P. aeruginosa* infection was evaluated in the animal model *G. mellonella*. As shown in Fig. 5, *G. mellonella* larvae infected with PA14 strain showed 75% and 100% mortality, respectively, after 18 and 20 h of inoculation. Treatment of PA14-infected larvae with ZC01 phage resulted in survival

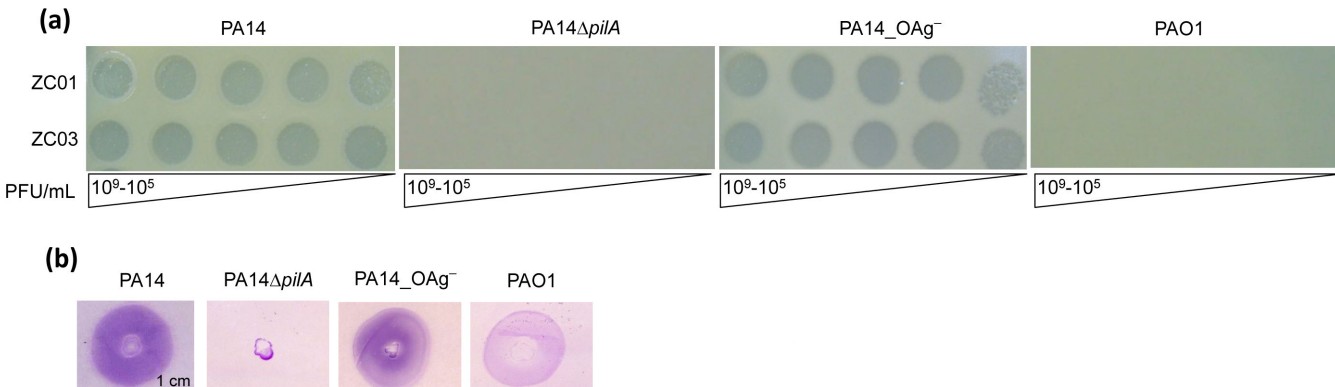

**FIG 2** Infection of *P. aeruginosa* PA14 depends on T4P but not on LPS. (a) ZC01 and ZC03 phage lysates at the indicated titers ($10^9$–$10^5$ PFU/mL) were spotted on wild-type strains PAO1 and PA14 and on T4P (PA14Δ*pilA*) and LPS (PA14_OAg⁻) null mutants. Photographs were taken after overnight incubation at 37°C. (b) Twitching motility assays of wild-type and mutant strains were analyzed by staining the plates with 0.1% crystal violet after 48 h. Images show a representative experiment of at least three independent biological replicates.

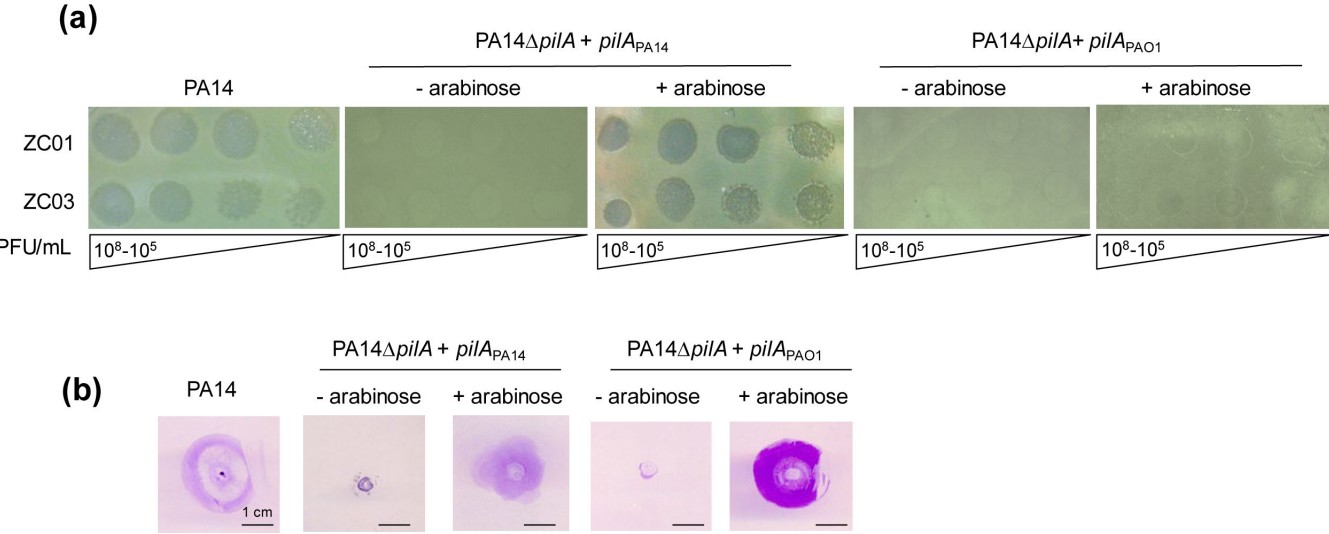

**FIG 3** Expression of PilA$_{PA14}$ restores phage-susceptibility of the PA14Δ*pilA* mutant. (a) ZC01 and ZC03 phage lysates at the indicated titers ($10^8$–$10^5$ PFU/mL) were spotted on PA14 and on PA14Δ*pilA* expressing *pilA*$_{PA14}$ or *pilA*$_{PAO1}$ upon induction with arabinose. Photographs were taken after overnight incubation at 37°C. (b) Twitching motility assays of wild-type and complemented strains induced or not with arabinose were analyzed by staining the plates with 0.1% crystal violet after 48 h. Images show a representative experiment of at least three independent biological replicates.

rates at 20 h of 53% and 68% at MOI of 20 and 100, respectively (Fig. 5a). After 24 h, ZC01 treatment at both MOIs resulted in an increase of 15%–21% ($P < 0.0001$) of the survival rate. Treatment with ZC03 resulted in survival rates of ~40% ($P < 0.0001$) at MOIs of 20 and 100, after 20–24 h post infection (Fig. 5b), although a dose-dependent effect was not observed as verified for ZC01. It should be mentioned that uninfected larvae inoculated with the phages or buffer alone presented 100% survival up to 24 h of the assay.

## DISCUSSION

In this work, we report an extended characterization of the phages ZC01 and ZC03 which were previously isolated from a thermophilic composting operation at the São

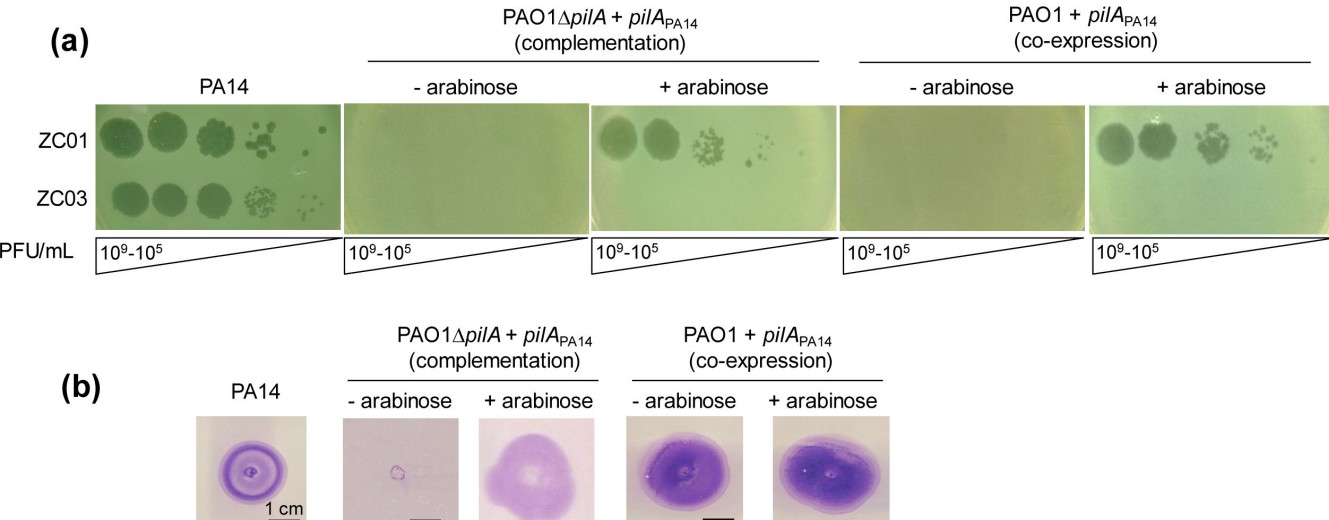

**FIG 4** Infection of *P. aeruginosa* PAO1 and PAO1 Δ*pilA* expressing PilA$_{PA14}$. (a) ZC01 and ZC03 phage lysates at the indicated titers ($10^9$–$10^5$ PFU/mL) were spotted on PAO1 or PAO1Δ*pilA* expressing PilA$_{PA14}$ upon induction using arabinose. Photographs were taken after overnight incubation at 37°C. (b) Twitching motility assays of wild-type and complemented strains induced or not with arabinose were analyzed by staining the plates with 0.1% crystal violet after 48 h. Images show a representative experiment of at least three independent biological replicates.

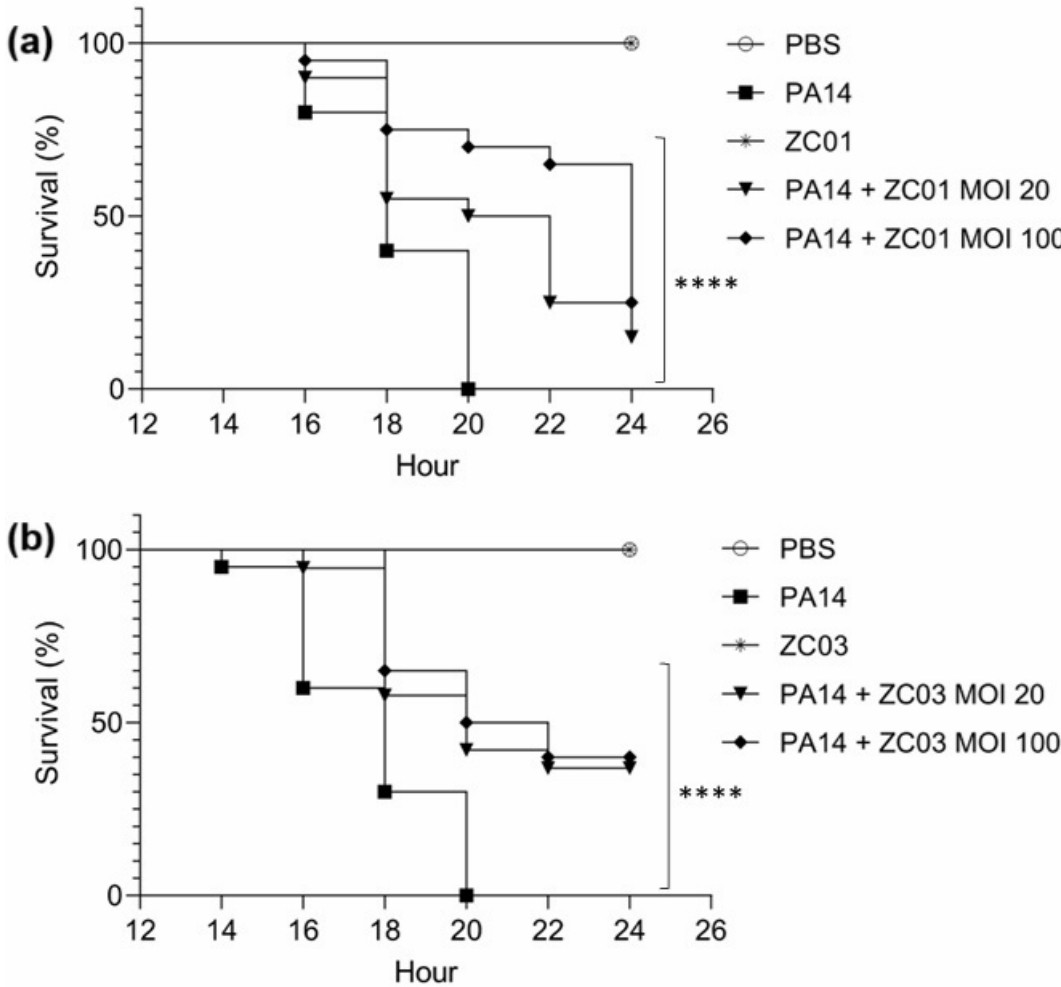

**FIG 5** *In vivo* efficacy of phages ZC01 and ZC03 against *P. aeruginosa* PA14 strain in *G. mellonella* infection model. Survival curves of *G. mellonella* larvae treated with ZC01 (a) or ZC03 (b). *G. mellonella* larvae were injected with buffer-only [phosphate-buffered saline (PBS)], PA14-only ($5 \times 10^3$ CFU/mL), phage-only ($10^5$ PFU/mL), and phage at MOI of 20 ($10^5$ PFU/mL) or 100 ($5 \times 10^5$ PFU/mL) 1 h post-infection with PA14 ($5 \times 10^3$ CFU/mL). The larvae were monitored at 2 h intervals, for 24 h. The data represent three independent experiments with 20 animals per treatment. Log rank (Mantel-Cox) test (****, $P < 0.0001$).

Paulo Zoo Park (Brazil) using *P. aeruginosa* PA14 as the enrichment host (46). ZC01 is a siphovirus currently classified within the *Abidjanvirus* genus, and ZC03 is a podovirus that belongs to the *Zicotriavirus* genus of the *Schitoviridae* N4-like viruses. These phages are devoid of any known lysogenic, virulence, or toxin genes that would preclude their use in phage therapy. Nevertheless, both genomes encode several ORFs of unknown function which require further characterization to verify that they are not harmful as therapeutic phages. Interestingly, although phages ZC01 and ZC03 were isolated from a thermophilic compost, they are not thermostable but maintain viability at 37°C and pH 7.5, which is considered satisfactory for therapeutic phages (56).

Besides the typical structural proteins of *Caudoviricetes*, proteins of unknown function (hypothetical proteins) were reliably identified by proteomics as components of ZC01 and ZC03 viral particles. It is worth noting that ZC01_056 and ZC01_073 amino acid sequences are conserved (>93% coverage and >57% identity) in other phages from the *Abidjanvirus* genus. On the other hand, some of the hypothetical proteins identified in the ZC03 proteome (ZC03_001, ZC03_003, ZC03_004, ZC03_008, ZC03 _014, ZC03_029, and ZC03_044) have similar counterparts only in *Pseudomonas* phage ZC08 (46) which together with ZC03 are the unique members of *Zicotriavirus* genus. Further analysis of the ZC03 proteome revealed hypothetical proteins ZC03_017 and ZC03_018, which

exhibit similarity to counterparts in other phages. Conversely, proteins ZC03_028 and ZC03_095 appear to be unique to phage ZC03. These observations support a change in the current annotation of these hypothetical proteins to virion structural proteins. We highlight ORFs ZC03_005 and ZC03_006 predicted as tail fiber proteins in ZC03 proteome which do not share amino acid sequence similarity. It has yet to be investigated whether ZC03_005 and ZC03_006 make up distinct tail fibers that could function as distinct RBPs, or whether both together are components of the ZC03 tail fibers.

Among the proteins identified in the ZC03 virion proteome, there is an 831 amino acids-protein (ZC03_016) which has an almost identical ortholog (100% coverage and 96% identity) only in *Pseudomonas* phage ZC08 (NCBI Reference Sequence accession NC_048639). ZC03_016 contains a lytic transglycosylase domain that belongs to the lysozyme-like domain superfamily predicted as peptidoglycan hydrolase according to InterPro classification (57). Virion-associated peptidoglycan hydrolase (VAPGH) proteins are generally attached to the viral particle contacting the bacterial surface in the first step of the infection process to locally degrade the bacterial cell wall peptidoglycan (58, 59). These enzymes have been proposed as antimicrobial and biotechnological tools to fight against numerous pathogens (56, 59–61). Thus, the predicted VAPGH domain in ZC03_016 can be further explored as an antimicrobial against *P. aeruginosa* and other Gram-negative pathogens.

By using a new set of 66 *P. aeruginosa* clinical isolates, we have confirmed the narrow and relatively distinct host range of phages ZC01 and ZC03. Despite this restricted host range, for a few clinical isolates, moderate-to-high EOP values were observed. According to predictions based on their genome sequences, these isolates present distinct sequence types as well as distinct serotypes from the PA14 or PAO1 strains (data not shown). Further genomic analyses may help to explain the moderate susceptibility of these strains to phages ZC01 and ZC03. While the host range of phages has traditionally been linked to receptor-associated properties (2), recent research has highlighted the significance of defense system quantity in determining *P. aeruginosa* strains susceptibility to phages (62).

As reported for other phages infecting *P. aeruginosa* (9–11, 18, 63), T4P is the primary receptor for phages ZC01 and ZC03 because they did not lyse a T4P-less host mutant (PA14Δ*pilA*). Phage adsorption along the pilus length occurs probably by their binding to PilA (the major component of T4P) (64). While T4P is a well-established receptor for phage adsorption, the exact mode of phage binding has not been clarified. Alterations in PilA sequence or its glycosylation can interfere with phage infection (18). Here, we provide additional evidence that PilA is indeed the T4P component recognized by ZC01 and ZC03 and that variations in pilin A sequence influence host recognition. Complementation of the T4P-less host mutant (PA14Δ*pilA*) with PilA$_{PAO1}$ cannot restore susceptibility to infections of phages ZC01 and ZC03. Nevertheless, PilA$_{PA14}$ expressed in PAO1Δ*pilA* turns this strain susceptible to infection by phage ZC01, but not to ZC03. From these observations, we can conclude that PAO1 does not carry anti-phage defense systems to abolish ZC01 infection as this phage can be replicated in PAO1 expressing PilA$_{PA14}$.

The opposite situation was observed for phage ZC03, which did not infect PAO1 expressing PilA$_{PA14}$, even though its T4P is functional. Thus, although pilin A sequence variation may be the primary host barrier for phage susceptibility, it is not sufficient to overcome PAO1 resistance to ZC03 infection. We can foresee some possible explanations for this result, such as that ZC03 adsorption is impaired by modifications of PA14 pilin A such as O-glycosylation (18) when it is expressed in PAO1. Another possible explanation could be that a surface structure, such as LPS, prevents ZC03 interaction with PAO1 by masking the host receptor. For instance, PAO1 LPS (O5 serotype) has an O-antigen composition quite distinct of PA14 LPS (serotype O19) (13, 65). Alternatively, ZC03 adsorption may depend on a second receptor recognized only in PA14 cells. Moreover, we cannot exclude that PAO1 carries anti-phage defense systems that impair ZC03 replication.

Phages ZC01 and ZC03 showed promising efficacy to treat PA14 infection in *G. mellonella* larvae as single-dose treatments with phages ZC01 or ZC03 significantly increase the survival of larvae infected with PA14 strain. It is worth mentioning that these phages exhibit a narrow host range and target a receptor (PilA) that is highly variable among clinical isolates (66, 67), which can be seen as drawbacks of using these phages in therapeutic applications (68). Nevertheless, our results warrant further studies to explore other dose regimens and/or the treatment with these phages together in a cocktail with other phages of different host ranges and/or targeting different receptors in larvae infected with PA14 or other susceptible clinical strains. Our work also lays the groundwork for a more detailed investigation of phage-bacteria-specific recognition mechanisms, especially considering ZC03, which is so far the only representative of the *Zicotriavirus* genus.

## MATERIALS AND METHODS

### *P. aeruginosa* strains and culture conditions

*P. aeruginosa* reference strains PA14 (69) and PAO1 (70), as well as clinical and environmental *P. aeruginosa* isolates (Table S3), were grown overnight at 37°C in TSB (tryptic soy broth) containing or not 1.5% agar (TSB-agar). Bacterial stocks were maintained in TSB supplemented with 10% glycerol at −80°C or in liquid nitrogen storage tank.

### Propagation and purification of phages ZC01 and ZC03

Previously isolated phages ZC01 and ZC03 (46) were propagated in *P. aeruginosa* PA14 using the double-layer agar technique (71) and purified following formerly described protocols (72, 73). Briefly, 0.5 mL of log-phase bacterial culture was mixed with the phage suspension (~50 µL) at an MOI of 0.01 and incubated for 10 min at 37°C. The phage-bacteria suspension was then mixed with 13 mL of molten TSB containing 0.7% agar (TSB-top-agar 0.7%) and then poured on a TSB-agar Petri dish. After overnight incubation at 37°C, 15 mL of Saline-Magnesium (SM) buffer (3% NaCl, 10 mM MgSO$_4$, 10 mM CaCl$_2$, 30 mM Tris-HCl, and pH 7.5) was added, and after gently shaking for 1 h at room temperature, the Petri dishes were incubated at 4°C overnight and for additional 1 h at room temperature without shaking. The lysates obtained from each plate were pooled, transferred to 50 mL conical tubes, and centrifuged at 6,000 × $g$ for 20 min at 4°C. The supernatants were transferred to clean tubes, and chloroform was added to the final concentration of 10% and immediately centrifuged at 6,000 × $g$ for 20 min at 4°C. The supernatant was filtered through a 0.22 µm membrane and titrated to calculate phages forming units per mL (PFU/mL) . A solution of 2.5 M NaCl in 20% polyethylene glycol 8000 was added to the filtered phage suspension in a 4:1 vol ratio. After brief mixing, the mixture was kept at 4°C for 24 h, and phage particles were pelleted by centrifugation (6,000 × $g$, 20 min, 4°C). The supernatant was then discarded, a new round of centrifugation was done, and the pellet with no traces of the supernatant was resuspended in SM buffer overnight at 4°C. The phages were further purified by cesium chloride (CsCl) density gradient ultracentrifugation (40,000 × $g$, 4 h, 4°C). Phages concentrated in a single band in the CsCl gradient were collected with a needle and syringe and subjected to dialysis by centrifugation through Amicon Ultra-15 Centrifugal Filters 100 kDa with two washes of SM buffer (5× the collected volume). Purified phages were titrated using a double-layer agar technique and stored at 4°C in SM buffer. Phages stocks were maintained in SM buffer supplemented with 10% glycerol at −80°C.

### Transmission electron microscopy

Around 3 µL of purified phage suspension was gently placed on glow-discharged carbon-coated 300 mesh copper grids. After about 1 min, excess liquid was blotted off, and the grid was stained with 2% uranyl acetate and air-dried. The negatively stained

phage particles were visualized with a JEOL JEM 2100 transmission electron microscope (JEOL Ltd, Tokyo, Japan) at operating voltage of 100 kV, and the images were registered digitally according to protocols of the Chemistry Institute Analytical Center (CA-USP, https://ca2.iq.usp.br/).

## Spot test and EOP assays

For the spot test assay, 120 µL of log-phase *P. aeruginosa* culture was mixed with 4 mL of molten TSB top-agar 0.7% and poured onto a TSB-agar Petri dish. After solidification, 4 µL of 10-fold serial dilutions (four dilutions) of the phage suspension was gently dropped over the top agar and examined for the presence of lysis plaques after overnight incubation at 37℃.

For the EOP assay, an overnight culture of phage susceptible *P. aeruginosa* strain was diluted to $OD_{600\ nm}$ = 1.0 (~3 × $10^9$ colony forming units per mL or CFU/mL), incubated for 10 min at 37℃ with 10 µL of 10-fold serial dilutions (eight dilutions) prepared from phage stocks at $10^{13}$–$10^{16}$ PFU/mL. The phage-bacteria suspension was mixed with 7 mL of molten TSB-top-agar 0.7%, poured on a 90 mm-TSB-agar Petri dish, and incubated overnight at 37℃. The EOP (74) was calculated by dividing the number of lysis plaques produced in each susceptible strain (for a fixed dose of phages) by the number of plaques produced in the host strain *P. aeruginosa* strain PA14. EOP values >0.1% and >50% are considered moderate or high, respectively (75).

## One-step growth curve

A total of 5 mL of a log-phase *P. aeruginosa* PA14 culture grown in TSB at 37℃ was mixed with 5 mL of a phage suspension to reach an MOI < 0.1. After incubation for 10 min at 37℃, the mixture was centrifuged at 6,000 × *g* for 15 min, and the supernatant was collected and titrated to determine the amount of phage that did not adsorb onto the bacteria. The pellet was resuspended in 20 mL of TSB and incubated at 37℃ without shaking. Every 10 min up to 190 min of incubation, 1 mL was collected and immediately titrated by the double-layer agar method (71) by plating 10-fold serial dilutions with *P. aeruginosa* PA14 and determining the number of lysis plaques produced (PFU/mL).

## Time-killing curve

To investigate the antibacterial effect of phages ZC01 and ZC03, overnight *P. aeruginosa* PA14 cultures were diluted to $10^6$ CFU/mL, which is equivalent to $OD_{600nm}$ of 0.2, and infected with phage at different MOIs (100, 10, 1, 0.1, and 0.01). The control sample had no phages. One hundred fifty microliter of each mixture was transferred to six wells of a 96-well plate and incubated at 37℃ for 12 h. $OD_{600nm}$ changes were measured every 15 min of incubation using the SpectraMax Paradigm microplate reader (Molecular Devices, CA, USA).

## Phage stability evaluation

To evaluate the effect of pH on phage stability, 100 µL of the phage suspension (1 × $10^9$ PFU/mL) prepared in SM buffer (pH 7.5) was mixed with 900 µL of universal buffer solution (150 mM KCl, 10 mM $KH_2PO_4$, 10 mM sodium citrate, and 10 mM $H_3BO_3$) at pHs 2, 4, 7.5, 9, and 12, followed by incubation for 60 min at 37℃. As a control, the phages were mixed with SM buffer (pH 7.5). The effect of temperature on phage stability was evaluated by incubation of 50 µL of the phage suspension (1 × $10^8$ PFU/mL) at 16℃, 25℃, 37℃, 42℃, 60℃, and 80℃ in a thermal cycler for 60 min. To evaluate the effect of chloroform, the phage suspension (1 × $10^9$ PFU/mL) was mixed with chloroform to a final concentration of 10% (vol/vol) and incubated for 60 min at room temperature. To determine the phage tolerance to UV light, 100 µL of a phage suspension was exposed to UV light (254 nm) of a germicidal lamp in a biosafety cabinet for 20 min. After the

treatments, the samples were immediately titrated by the double-layer agar method (71) by plating 10-fold serial dilutions with *P. aeruginosa* PA14 and determining PFU/mL.

## Mass spectrometry-based proteomics

The identification of peptides of pure preparations of ZC01 and ZC03 phage particles was performed using two methods of mass spectrometry-based proteomics (SDS-PAGE followed by in-gel digestion and in-solution digestion). In the first approach, the phage particle samples (~100 µg of total protein) were resuspended in 50 mM Tris pH 6.8, 25 mM DTT, 10% glycerol, 1% SDS, 0.025% bromophenol blue, and subjected to SDS-PAGE (76). The gel was stained with Coomassie blue G250, the bands were cut out (Fig. S3), washed with 50 mM $NH_4HCO_3$, 40% acetonitrile (ACN), and subjected to sequential incubations with 10 mM dithiothreitol (DTT), 100 mM iodoacetoamide, and ACN prior to trypsin digestion using 100 ng of sequencing grade trypsin and 10 mM $NH_4HCO_3$ for 16 h at 37°C. Digestion was stopped with 10% formic acid (FA), and the peptides were extracted with 40% ACN, 0.1% FA, concentrated by vacuum centrifugation, and desalted with Zip-Tip C18 Cartridge column. For in-solution digestion, the phage particles suspension (~100 µg of total protein) was dried by vacuum centrifugation and suspended in 8 M urea, 10 mM DTT, and protease inhibitors cocktail and was incubated for 30 min at 30°C. The samples were diluted 10 times to 0.8 M urea, and sequencing grade trypsin was added (1:50 enzyme:total protein) followed by incubation at room temperature for 16 h at 37°C. The digestion was stopped by adding 1% trifluoroacetic acid, and the samples were desalted with Zip-Tip C18 Cartridge column. Samples were then subjected to nanoflow liquid chromatography coupled to mass spectrometry at the BIOMASS Core Facility at the Center for Research Facilities (CEFAP-USP, https://cefap.icb.usp.br/) using an Easy-nLC system coupled to LTQ-Orbitrap Velos mass spectrometer (Thermo Fisher Scientific Inc., MA, USA). Samples were resuspended in 0.1% FA and loaded onto a C18 PicoFrit column [C18 PepMap, 75 µm id × 10 cm, 3.5 µm particle size, and 100 Å pore size (New Objective, Ringoes, NJ, USA)] and separated with a gradient from 100% mobile phase A (0.1% FA) to 34% phase B (0.1% FA, 95% ACN) during 60 min, at a flow rate of 300 nL/min. Samples were analyzed in duplicate. The LTQ-Orbitrap Velos was operated in positive polarity with data-dependent acquisition. The full scan was obtained in the Orbitrap at a resolution of 60,000 FWHM in the 350–1,500 m/z mass range. The 20 most abundant peptide ions obtained in the MS full scan were selected for MS/MS, fragmented using CID at 35 normalized collision energy, and dynamic excluded for 15 s. All raw data were assessed in the Xcalibur software (Thermo Fisher Scientific Inc., MA, USA). Tandem mass spectra were processed and searched against an in-house database composed of annotated ORFs in phages ZC01 and ZC03 genomes (NCBI Reference Sequence accessions NC_052965.1 and NC_048638.1) using Proteome Discovery v. 1.4 (Thermo Fisher Scientific Inc., MA, USA) and SEQUEST (77), with the following parameters: precursor mass tolerance of 10 ppm; MS/MS mass tolerance 0.6 Da (CID data). Trypsin was selected as enzyme, carbamidomethyl cysteine as fixed modification and oxidation of methionine as variable modification. The False Discovery Rates (FDRs) were calculated using the algorithm Percolator with equal or less than 0.01. Protein FDR was calculated in the Proteome Discoverer software and kept below 1%. The mass spectrometry proteomics data have been deposited to the ProteomeXchange via Consortium the PRIDE (78) partner repository with the data set identifier PXD055478 (https://www.ebi.ac.uk/pride/archive/).

## *P. aeruginosa* mutants and complementation of PA14ΔpilA and PAO1ΔpilA

The deletion mutants Δ*pilA* were constructed by allelic replacement and do not twitch (79). The spontaneous mutant PA14_OAg⁻ is part of our mutant collection and lacks the O-antigen structures of LPS (79). For complementation of the Δ*pilA* mutants, the *pilA* genes from PA14 and PAO1 strains were amplified by PCR using the following primer pairs: PilA_PA14_fwd_ccgtttttttgggctagcgTATCAATGGA-GAGATACATGAAAGCTC; PilA_PA14_rev_gcggccgctctagaactagtTTAGCGGCATTCGCTCGG

and PilA_PAO1_fwd_cccgttttttttgggctagcgATGAAAGCTCAAAAAGGC; PilA_PAO1_rev_gcggccgctctagaactagtTTAGTTATCACAACCTTTCG. The resulting PCR amplicons were cloned into pJN105 plasmid (80) using the one-step sequence and ligation-independent cloning (SLIC) protocol (81). The vector was linearized using inverse PCR with the following primer pair: pJN105_rev CGCTAGCCCAAAAAAACG; pJN105_fwd ACTAGTTCTAGAGCGGCC. Resulting SLIC constructs were transformed into *E. coli* DH5α BL21(DE3), and recombinant clones were selected with ampicillin (50 µg/mL). The authenticity of the cloned genes was verified by Sanger sequencing of the inserts. Constructs were introduced into the target *P. aeruginosa* strains by electroporation. Transformants were selected with gentamicin (50 µg/mL), and the expression of cloned *pilA* was induced with 0.2% L-arabinose.

## Twitching motility assays

Macroscopic twitching motility assays were performed by stabbing a single colony through a 3-mm-thick TSB-1% agar plate. After incubation at 37°C for 24 h in a humidified chamber, the agar was removed, and the twitching zone was stained for 15 min with 1% crystal violet. The stained area is proportional to the cells ability to twitch, which is dependent on T4P.

## Phage treatment of *G. mellonella* larvae infected with *P. aeruginosa*

The assays of *G. mellonella* phage treatment were based on previously described protocols (44, 82). Briefly, *G. mellonella* larvae with size ranging from 2.0 to 2.5 cm in length and body weight from 150 to 200 mg were surface sterilized with 70% ethanol, separated into groups of 20 larvae, and placed in polystyrene Petri dishes (140 mm of diameter). A 10 µL inoculum of *P. aeruginosa* PA14 ($5 \times 10^3$ CFU/mL) prepared in phosphate-buffered saline (PBS) was injected into the larva hemolymph behind the last proleg using a 10 µL Hamilton syringe. After 60 min, 10 µL of a phage suspension ($1 \times 10^5$ or $5 \times 10^5$ PFU/mL) or PBS (positive control group) was delivered behind the last proleg on the opposite site to the bacterial injection site. Negative control groups (one group injected with PBS only to assess the impact of any negative effect from the injection process, and one group injected with phage suspension only to assess toxicity of the phage suspension) were also included. The larvae were kept at 37°C fed with pollen and beeswax. After 13 h post-infection, the larvae survival on each group was monitored every 2 h, for 12 h. The larvae were recorded as dead when they did not move in response to touch. Kaplan–Meier survival curves and log-rank (Mantel–Cox) statistical test were performed using GraphPad Prism 10.2.2 (GraphPadSoftware LLC.).

## ACKNOWLEDGMENTS

The authors are grateful to the members of the Biology of Bacteria and Bacteriophages Research Center (CEPID B3) for discussions and to Drs. Ana Cristina Gales, Ana Lúcia da Costa Darini, Cintya de Oliveira Souza, Karla Lima, and Nilton Erbet Lincopan Huenuman for providing various *P. aeruginosa* isolates for host range assays.

Funding for this work was provided by the São Paulo Research Foundation (FAPESP; grant numbers 2021/10577–0, 2018/18257–1, 2018/15549–1, 2020/04923–0, and 2022/11334–6) and the National Council for Scientific and Technological Development (CNPq; grant number 314701/2020–6). A.P.S.J. is supported by a PhD fellowship from the Coordination for the Improvement of Higher Education Personnel (CAPES). A.M.D.S., G.P., and K.I. are supported in part by Research Fellowship Awards from CNPq. The funders had no role in study design, data collection, analysis, decision to publish, or preparation of the manuscript.

L.F.M. and A.M.D.S. designed the research. L.F.M. and A.P.S.J. conducted the experiments. R.B.Q. and G.G.S. contributed to the large-scale phage purification and transmission electron microscopy. C.B.A. and G.P. supplied laboratory infrastructure and performed proteomic analysis. G.G.N. and R.L.B. generated *P. aeruginosa* PA14 deletion mutants. G.S. and K.I. supplied the laboratory infrastructure and assistance for *G.*

*mellonella* assays. L.F.M., A.P.S.J., F.P.N.R., G.G.S., R.L.B., and A.M.D.S. participated in the interpretation of the experiments and analysis of the data. L.F.M. and A.M.D.S. wrote the manuscript. All authors read, provided critical review, and approved the final manuscript.

## AUTHOR AFFILIATIONS

[1]Departamento de Bioquímica, Instituto de Química, Universidade de São Paulo, São Paulo, Brazil
[2]Biology of Bacteria and Bacteriophages Research Center (CEPID B3), São Paulo, Brazil
[3]Departamento de Microbiologia, Instituto de Ciências Biomédicas, Universidade de São Paulo, São Paulo, Brazil
[4]Departamento de Parasitologia, Instituto de Ciências Biomédicas, Universidade de São Paulo, São Paulo, Brazil
[5]School of Natural Sciences, Macquarie University, Sydney, Australia
[6]Departamento de Ciências BioMoleculares, Faculdade de Ciências Farmacêuticas de Ribeirão Preto, Universidade de São Paulo, São Paulo, Brazil

## AUTHOR ORCIDs

Layla Farage Martins  http://orcid.org/0000-0002-5277-8813
Kelly Ishida  http://orcid.org/0000-0002-4602-0926
Regina Lúcia Baldini  http://orcid.org/0000-0003-4349-6352
Aline Maria da Silva  http://orcid.org/0000-0001-9249-4922

## FUNDING

| Funder | Grant(s) | Author(s) |
|---|---|---|
| Fundação de Amparo à Pesquisa do Estado de São Paulo (FAPESP) | 2021/10577-0 | Germán Gustavo Sgro<br>Regina Lúcia Baldini<br>Aline Maria da Silva |
| Fundação de Amparo à Pesquisa do Estado de São Paulo (FAPESP) | 2018/18257-1 | Giuseppe Palmisano |
| Fundação de Amparo à Pesquisa do Estado de São Paulo (FAPESP) | 2018/15549-1 | Giuseppe Palmisano |
| Fundação de Amparo à Pesquisa do Estado de São Paulo (FAPESP) | 2020/04923-0 | Giuseppe Palmisano |
| Fundação de Amparo à Pesquisa do Estado de São Paulo (FAPESP) | 2022/11334-6 | Giuseppe Palmisano |
| Conselho Nacional de Desenvolvimento Científico e Tecnológico (CNPq) | 314701/2020-6 | Aline Maria da Silva |
| Coordenação de Aperfeiçoamento de Pessoal de Nível Superior (CAPES) | | Ariosvaldo Pereira dos Santos Junior |

## AUTHOR CONTRIBUTIONS

Layla Farage Martins, Conceptualization, Data curation, Formal analysis, Investigation, Methodology, Writing – original draft, Writing – review and editing | Ariosvaldo Pereira dos Santos Junior, Data curation, Formal analysis, Investigation, Methodology | Gianlucca Gonçalves Nicastro, Investigation, Methodology, Resources | Gaby Scheunemann, Formal analysis, Investigation, Methodology, Resources | Claudia Blanes Angeli, Data curation, Investigation, Methodology, Resources | Fernando Pacheco Nobre Rossi, Data curation, Methodology, Software | Ronaldo Bento Quaggio, Formal analysis, Investigation, Methodology | Giuseppe Palmisano, Data curation, Formal analysis, Funding acquisition, Investigation, Methodology, Resources, Writing – review and editing | Germán Gustavo Sgro, Data curation, Formal analysis, Methodology, Writing – review and editing | Kelly

Ishida, Formal analysis, Methodology, Resources | Regina Lúcia Baldini, Formal analysis, Investigation, Methodology, Resources | Aline Maria da Silva, Conceptualization, Data curation, Formal analysis, Funding acquisition, Project administration, Resources, Supervision, Visualization, Writing – original draft, Writing – review and editing

## ADDITIONAL FILES

The following material is available online.

### Supplemental Material

**Fig. S1 (Spectrum01527-24-s0001.pdf).** One-step growth curve for phage ZC01.
**Fig. S2 (Spectrum01527-24-s0002.pdf).** Time-killing curves of *P. aeruginosa* PA14 exposed to phages ZC01 and ZC03.
**Fig. S3 (Spectrum01527-24-s0003.pdf).** ZC01 and ZC03 phage particles subjected to SDS-PAGE.
**Supplemental tables (Spectrum01527-24-s0004.xlsx).** Tables S1 and S2.
**Table S3 (Spectrum01527-24-s0005.xlsx).** Host range of phages ZC01 and ZC03 on 69 clinical or environmental *P. aeruginosa* isolates.

### Open Peer Review

**PEER REVIEW HISTORY (review-history.pdf).** An accounting of the reviewer comments and feedback.

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
