## [Reviewer comments · Microbiology Spectrum]

Microbiology Spectrum

Phages ZC01 and ZC03 require Type-IV Pilus for *Pseudomonas aeruginosa* infection and have a potential for therapeutic applications

Layla Martins, Ariosvaldo dos Santos Junior, Gianluca Nicastro, Gaby Scheunemann, Claudia Angeli, Fernando Rossi, Ronaldo Quaggio, Giuseppe Palmisano, Germán Sgro, Kelly Ishida, Regina Baldini, and Aline Maria da Silva

Corresponding Author(s): Aline Maria da Silva, Universidade de Sao Paulo

Review Timeline:

Submission Date:	July 22, 2024
Editorial Decision:	August 19, 2024
Revision Received:	September 26, 2024
Accepted:	September 30, 2024

Editor: Ethel Bayer-Santos

Reviewer(s): The reviewers have opted to remain anonymous.

Transaction Report:

DOI: <https://doi.org/10.1128/spectrum.01527-24>

Re: Spectrum01527-24 (Phages ZC01 and ZC03 require Type-IV Pilus for *Pseudomonas aeruginosa* infection and have a potential for therapeutic applications)

Dear Prof. Aline Maria da Silva:

Thank you for the privilege of reviewing your work. Below you will find instructions from the Spectrum editorial office, and the reviewer comments.

Revision Guidelines

Sincerely,
Ethel Bayer-Santos
Editor
Microbiology Spectrum

Reviewer #1 (Public repository details (Required)):

proteomics dataset

Reviewer #1 (Comments for the Author):

Here the authors investigate phage microbe interactions, that includes characterization of phage proteomes, the identification of the receptor of the phages (the T4P of *P. aeruginosa*) and the utility of using these phages to mitigate the impact of a *P.*

aeruginosa infection in an insect infection model. The focus of this work is phages ZC01 and ZC03 - the isolation and sequencing of the phage genomes was reported previously. The proteomics studies and re-analysis of the genomes has helped to identify putative functions for previously hypothetical proteins. The authors also show the phages can infect other *P. aeruginosa* isolates.

This is a solid manuscript and was enjoyable to read. I have only a few additional comments/clarifications below.

Specific comments

1. I believe the "Impact Statement" should be renamed "Importance".
2. Lines 85-90. Another phage of *Pa* known to bind the T4P is DMS3.
3. Line 103. I believe only PA14 comes from a burn. PAO1 comes from a wound (<https://www.microbiologyresearch.org/content/journal/micro/10.1099/00221287-13-3-572>)
4. There is a long discussion of phage proteins driving binding to the host cells, but this point is not really explored in the manuscript, so not sure it is needed in Intro.
5. I will have to defer to a taxonomy expert as to the classification of these phages (which will likely change again next week).
6. It would be helpful to readers to mention in the abstract that these are lytic phages.
7. The studies of the phage looking at various clinical strains (conferring narrow host range) and identifying T4P as the likely receptor are solid and well-controlled.
8. The conclusion drawn from the expt in Figure 3 is reasonable given that the PilAPO1 can complement the twitching phenotype of the PA14 pilA mutant. This very nice control tells us that the PilAPO1 protein is expression and functional in PA14. This finding is nicely confirmed with the data in Figure 4.
9. The data in Figure 5 are solid and well controlled.

Reviewer #2 (Comments for the Author):

This is a straightforward paper on the characterization of two *P. aeruginosa* phages. The paper is well written and provides new insights into the receptors and host range for these phages. I only have minor comments:

I think the discussion could be shortened, as it largely restates the results. The first paragraph could also be removed. However, it would be valuable to discuss:

1. The potential drawbacks to these phages, mainly that they have limited host range, target a receptor that is often times not expressed in *P. aeruginosa* chronic infections, and is often mutated in chronic isolates.
2. Are the genome sequences available for the *P. aeruginosa* isolates tested? If so, do they have intact genes for Type IV pili production and function? Does the pilA sequence for susceptible strains resemble PAO1 or PA14?
3. The complementation experiments with heterologous PilA are convincing, yet it doesn't appear that they mirror what was observed (in regard to the degree of twitching motility complementation) by Asikyan and Burrows (J Bact 2008). Can you discuss these data?

Line 92: The authors may want to change ESKAPE to ESKAPEE

Line 144: should be latent 'period'

I would suggest changing the name of PA14_LPS- to PA14_OAg- to be more specific. A naïve reader may think that this strain has no LPS components, since this has been accomplished in another Gram negative (*Acinetobacter*).

Any reason why you think the 100 MOI of ZC03 didn't protect better than the MOI 20? Seems it did work better for ZC01.

Table 1 legend: should be titer not title

Overall, well done paper.

Point-by-point responses to reviewers' comments on the manuscript Spectrum01527-24

Manuscript: Phages ZC01 and ZC03 require Type-IV Pilus for *Pseudomonas aeruginosa* infection and have a potential for therapeutic applications by Martins *et al.*

We thank both reviewers for the careful reading of our manuscript and their constructive comments and suggestions. We have considered all the suggestions in our revised manuscript. Our responses are in what follows. In italics are the comments by the Reviewer and our responses are in normal font. All changes in the manuscript are highlighted in yellow in the manuscript marked up file.

Reviewer #1:

Comments to the Author

General comments:

Reviewer #1 (Public repository details (Required)):

proteomics dataset

R: The mass spectrometry proteomics data have been deposited to the ProteomeXchange Consortium via the PRIDE partner repository with the dataset identifier PXD055478. Data will be publicly available upon publication of the manuscript and can be accessed by the reviewer with following account details [Username:reviewer_pxd055478@ebi.ac.uk Password: nGJ9Hcdgg1Z2]. The information was included in the methods section of the revised manuscript (lines 457-459).

Reviewer #1 (Comments for the Author):

*Here the authors investigate phage microbe interactions, that includes characterization of phage proteomes, the identification of the receptor of the phages (the T4P of *P. aeruginosa*) and the utility of using these phages to mitigate the impact of a *P. aeruginosa* infection in an insect infection model. The focus of this work is phages ZC01 and ZC03 - the isolation and sequencing of the phage genomes was reported previously. The proteomics studies and re-analysis of the genomes has helped to identify putative functions for previously hypothetical proteins. The authors also show the phages can infect other *P. aeruginosa* isolates.*

This is a solid manuscript and was enjoyable to read. I have only a few additional comments/clarifications below.

R: We thank the reviewer for appreciating our manuscript and the soundness of our results. We are also grateful for the reviewer's comments, which have helped us to produce a much-improved version of the manuscript. Responses to the reviewer's comments follow below.

Specific comments

1. I believe the "Impact Statement" should be renamed "Importance".

R: The section was renamed as suggested. (line 53 in the revised text)

2. Lines 85-90. Another phage of Pa known to bind the T4P is DMS3.

R: The reference regarding DMS3, another type IV pilus-dependent phage, was included in the revised manuscript. [Budzik JM, Rosche WA, Rietsch A, O'Toole GA. 2004. Isolation and characterization of a generalized transducing phage for *Pseudomonas aeruginosa* strains PAO1 and PA14. J Bacteriol 186:3270-3.] (lines 83; 600-602)

3. *Line 103. I believe only PA14 comes from a burn. PAO1 comes from a wound*
(<https://www.microbiologyresearch.org/content/journal/micro/10.1099/00221287-13-3-572>)

R: The reviewer is correct. The text and references were revised as follows: “PAO1 and PA14 (or UCBPP-PA14) are commonly used as laboratory reference *P. aeruginosa* strains to study this bacterial species (38-41). While PAO1 is a derivative of the original PAO isolate which was obtained from a wound at Melbourne, PA14 was one of the strains isolated from burn wound patients at a hospital in Pennsylvania (38, 39, 42).” (lines 100-103; 690-691)

4. *There is a long discussion of phage proteins driving binding to the host cells, but this point is not really explored in the manuscript, so not sure it is needed in Intro.*

R: We agree with the reviewer and have removed the last two sentences in this paragraph: “RBP amino acid sequences are highly diverse, which can hamper their identification when analyzing phage genomes (7, 9). Nevertheless, identification of putative RBPs on phage genomes can be undertaken by conserved domain analyses and structure predictions (6, 10)”. Please see lines 72-80 in the revised text.

5. *I will have to defer to a taxonomy expert as to the classification of these phages (which will likely change again next week).*

R: We agree with the reviewer that phage taxonomy may change in the foreseeable future. For this work, we have followed the current taxonomic classification for phages based on Turner D et al. 2023. Abolishment of morphology-based taxa and change to binomial species names: 2022 taxonomy update of the ICTV bacterial viruses subcommittee. Archives of Virology 168:74.

6. *It would be helpful to readers to mention in the abstract that these are lytic phages.*

R: The mention that ZC01 and ZC03 are lytic phages was included in the abstract (line 31)

7. *The studies of the phage looking at various clinical strains (conferring narrow host range) and identifying T4P as the likely receptor are solid and well-controlled.*

8. *The conclusion drawn from the expt in Figure 3 is reasonable given that the PilAPO1 can complement the twitching phenotype of the PA14 delta pilA mutant. This very nice control tells us that the PilAPO1 protein is expressed and functional in PA14. This finding is nicely confirmed with the data in Figure 4.*

9. *The data in Figure 5 are solid and well controlled.*

R: We thank the reviewer for recognizing the soundness of our results.

Reviewer #2:**Comments to the Author**

*This is a straightforward paper on the characterization of two *P. aeruginosa* phages. The paper is well written and provides new insights into the receptors and host range for these phages. I only have minor comments:*

R: We thank the reviewer for appreciating our manuscript and new insights gained from our results. We

are also grateful for the reviewer's comments, which have helped us to produce a much-improved version of the manuscript. Responses to the reviewer's comments follow below.

I think the discussion could be shortened, as it largely restates the results. The first paragraph could also be removed. However, it would be valuable to discuss:

R: We agree with the reviewer. The discussion was revised to avoid restating the results (tables and figures), and the first paragraph was removed as suggested. Please see the revised text lines 236, 248-249, 259-260, 277.

1. The potential drawbacks to these phages, mainly that they have limited host range, target a receptor that is often times not expressed in P. aeruginosa chronic infections, and is often mutated in chronic isolates.

R: Following the reviewer's suggestion, we have modified the discussion to address the potential drawbacks of ZC01 and ZC03. Please see the revised text lines 315-320.

2. Are the genome sequences available for the P. aeruginosa isolates tested? If so, do they have intact genes for Type IV pili production and function? Does the pilA sequence for susceptible strains resemble PAO1 or PA14?

R: Unfortunately, the genome sequences for **all** the *P. aeruginosa* isolates tested in our work are not currently available. We did have sequenced the genomes for 16 clinical isolates (3845 GSP-3 producer; ALERTA 226/GES-5 producer; ALERTA 275/ VIM-7 producer; ALERTA 395/IMP-18 producer; Fc79a M; Fc79a PAB NM; Fc7f NM; MT138; MT222; P13.612; SC-116; SC-61; SC-84; H6044; H6086; 5757 - susceptible isolates to both phages on drop test assay are underlined) and resequenced the genomes of PA14 and PAO1 reference strains. The genes for type IV pili production and function are intact in the clinical isolates and, as expected, in PA14 and PAO1. In addition, they all have functional type IV pilus according to twitching motility assays (data not shown). The genome sequences and phenotypes for these 16 clinical isolates will be soon publicly available as part of another ongoing work.

Alignment of the pilA amino acid sequences of these isolates does reveal differences (please see the MAFFT alignment and sequence identity comparison below). However, it was not possible to detect a clear correlation of such differences with susceptibility to the phages ZC01 or ZC03. For instance, the PilA of isolates 5757 and Fc79a PAB NM is more similar to PilA of PAO1 than to PA14 PilA, and these three isolates show moderate susceptibility to phages ZC01 and ZC03, in contrast to PAO1 which is fully resistant.

We have chosen not to discuss these observations in the present manuscript since they do not clearly explain the susceptibility or resistance of the isolates to phages. Other factors (PilA glycosylation, phage immune system) are certainly combined to explain the narrow host range observed for these phages.

3. The complementation experiments with heterologous PilA are convincing, yet it doesn't appear that they mirror what was observed (in regard to the degree of twitching motility complementation) by Asikyan and Burrows (J Bact 2008). Can you discuss these data?

R: Possible explanations for the difference between our results and those reported by Asikyan et al 2008 [Asikyan ML, Kus JV, Burrows LL. 2008. Novel proteins that modulate type IV pilus retraction dynamics in *Pseudomonas aeruginosa*. J Bacteriol 190:7022-34] could be related to different experimental conditions (growth media; complementation vector; PAO strain) of what we have used. Indeed, in a more recent article [Kim ES, Bae HW, Cho YH. 2018. A Pilin Region Affecting Host Range of the *Pseudomonas aeruginosa* RNA Phage, PP7. Frontiers in Microbiology 9:ARTN 247] it was observed that expression of the pilin A of PA14 (group III) and of PAK (group IIb) when expressed in PAO1 pilA mutant restored twitching motility, similarly what we have reported in the present work.

Line 92: The authors may want to change ESKAPE to ESKAPEE

R: Changed as suggested (lines 89, 91, 97 in the revised text). An additional reference was also included [Rajput A, Seif Y, Choudhary KS, Dalldorf C, Poudel S, Monk JM, Palsson BO. 2021. Pangenome Analytics Reveal Two-Component Systems as Conserved Targets in ESKAPEE Pathogens. mSystems 6:10.1128/msystems.00981-20.]

Line 144: should be latent 'period'

R: Corrected (line 143 in the revised text).

I would suggest changing the name of PA14_LPS- to PA14_OAg- to be more specific. A naïve reader may think that this strain has no LPS components, since this has been accomplished in another Gram negative (Acinetobacter).

R: Changed as suggested. Please see the revised text lines 199, 201, 204, 463 as well as Figure 2 and respective caption (line 541).

Any reason why you think the 100 MOI of ZC03 didn't protect better than the MOI 20? Seems it did work better for ZC01.

R: We thank the reviewer for commenting on this important issue. As commonly seen in the literature there is dose dependent effect with higher MOIs protecting better *G. mellonella* larvae than lower MOIs. On the other hand, this effect is not always observed and seems to be dependent of the phage and the bacterial host. For instance, in some works the authors did not observe protection significantly better with higher MOIs [Beeton ML, Alves DR, Enright MC, Jenkins ATA. 2015. Assessing phage therapy against *Pseudomonas aeruginosa* using a *Galleria mellonella* infection model. International Journal of Antimicrobial Agents 46:196-200]. Although we did not further explore this result, one possibility is that at higher concentrations ZC03 may aggregate, becoming unable to infect the bacterial host. Another possibility that high MOIs could lead to rapid bacterial resistance development [Międzybrodzki, 2023 #1092]. We have revised the text to address this issue (lines 231-232 in the revised text).

Table 1 legend: should be titer not title

R: Corrected. We apologize for this mistake.

Overall, well done paper.

R: We would again like to thank the reviewer for appreciating our manuscript.

Re: Spectrum01527-24R1 (Phages ZC01 and ZC03 require Type-IV Pilus for *Pseudomonas aeruginosa* infection and have a potential for therapeutic applications)

Dear Prof. Aline Maria da Silva:

Your manuscript has been accepted, and I am forwarding it to the ASM production staff for publication. Your paper will first be checked to make sure all elements meet the technical requirements. ASM staff will contact you if anything needs to be revised before copyediting and production can begin. Otherwise, you will be notified when your proofs are ready to be viewed.

Sincerely,
Ethel Bayer-Santos
Editor
Microbiology Spectrum